# Cell Penetrating Peptide as a High Safety Anti-Inflammation Ingredient for Cosmetic Applications

**DOI:** 10.3390/biom10010101

**Published:** 2020-01-07

**Authors:** Tse-Kai Fu, Ping-Hsueh Kuo, Yen-Chang Lu, Hsing-Ni Lin, Lily Hui-Ching Wang, Yu-Chun Lin, Yu-Chen Kao, Huey-Min Lai, Margaret Dah-Tsyr Chang

**Affiliations:** 1Institute of Molecular and Cellular Biology, National Tsing Hua University, Hsinchu 30013, Taiwan; gn00042366@gmail.com (T.-K.F.); s9980584nthu@gapp.nthu.edu.tw (P.-H.K.); johnnylu0629@gmail.com (Y.-C.L.); sisnin66@gamil.com (H.-N.L.); lilywang@life.nthu.edu.tw (L.H.-C.W.); 2Research and Development Department, Simpson Biotech Co., Ltd., Taoyuan 333, Taiwan; 3Department of Medical Science, National Tsing Hua University, Hsinchu 30013, Taiwan; 4Industrial Technology Research Institute, Hsinchu 30011, Taiwan; YuChunLin@itri.org.tw (Y.-C.L.); YCKao@itri.org.tw (Y.-C.K.); 5Department of Life Science, National Tsing Hua University, Hsinchu 30013, Taiwan

**Keywords:** cell-penetrating peptides, cosmeceutical peptides, anti-inflammation

## Abstract

Cosmeceutical peptides have become an important topic in recent decades in both academic and industrial fields. Many natural or synthetic peptides with different biological functions including anti-ageing, anti-oxidation, anti-infection and anti-pigmentation have been developed and commercialized. Current cosmeceutical peptides have already satisfied most market demand, remaining: “cargos carrying skin penetrating peptide with high safety” still an un-met need. To this aim, a cell-penetrating peptide, CPP_AIF_, which efficiently transported cargos into epithelial cells was exanimated. CPP_AIF_ was evaluated with cell model and 3D skin model following OECD guidelines without using animal models. As a highly stable peptide, CPP_AIF_ neither irritated nor sensitized skin, also did not disrupt skin barrier. In addition, such high safety peptide had anti-inflammation activity without allergic effect. Moreover, cargo carrying activity of CPP_AIF_ was assayed using HaCaT cell model and rapid CPP_AIF_ penetration was observed within 30 min. Finally, CPP_AIF_ possessed transepidermal activity in water in oil formulation without disruption of skin barrier. All evidences indicated that CPP_AIF_ was an ideal choice for skin penetrating and its anti-inflammatory activity could improve skin condition, which made CPP_AIF_ suitable and attractive for novel cosmeceutical product development.

## 1. Introduction

Peptides, polymers composed of amino acids, are known to possess versatile biological functions such as promoting cell proliferation, migration, inflammation/anti-inflammation, angiogenesis, and melanogenesis [1], which causes numerous physiological procedures in human body [2]. The first synthetic peptide incorporated into skin care products in the late 80s was copper glycine-histidine-lysine (Cu-GHK) generated by Pickard in 1973 [3]. Since then, many short synthetic peptides playing roles in inflammation, extracellular matrix synthesis or pigmentation have been developed. These peptides are used for anti-oxidation, whitening effects, “Botox-like” wrinkle smoothing and collagen stimulation.

Cosmeceutical peptides usually have certain features. Historically, it has always been assumed that the molecular weight of a peptide should be less than 500 Da, if not it could not pass skin barrier [4]. In addition, the peptide should have water solubility over 1 mg/mL and have no or few polar centers in its sequence [5]. Copper tripeptides, tetrapeptide PKEK, manganese tripeptide-1, soybean peptide, black rice oligopeptides and silk fibroin peptide have been on the market for several decades, but less in vivo efficacy data is available [5]. In general, their substance mixtures are examined in cosmetic formulations, so as to the actual effect of individual peptides on the skin still remains unclear in many cases.

Scientific research on peptides usually focus on identification of functional mechanism, and practical application for cosmetic and/or pharmaceutical use. In the last decade, the development of active peptides has established a new field in cosmeceutical and pharmaceutical skin care. To generate safety profile and functional data of peptide, in vivo animal test was usually chosen in the past. However, correctness of results from animal model for human skin has always been quested. This problem, using animals for testing purposes, finally leads to EU regulation (76/768/EEC, Feb. 2003), beginning in 2009, to prohibit use of animals to accumulating toxicological data for cosmetic ingredients.

As an alternative solution, artificial human skin models have been established and many of these are now commercially available. Several technologies have been introduced to design and develop artificial skin models which highly simulate complex structure of human skin [6,7]. The most common skin models were epidermis models using human skin epidermal cells, including EpiSkin^®^ (L’Oreal, Levallois-Perret, France), EpiDerm^®^ (MatTek Corporation, Ashland, MA, USA), SkinEthic^®^ (SkinEthics, Lyon, France) and epiCS^®^ (CellSystems, Troisdorf, Germany). Recently, some advanced skin models were commercialized, including Phenion^®^ (Henkel, Düsseldorf, Germany) and NeoDerm^®^ (Tegoscience, Seoul, Korea). These models have been proven to replace animal tests in pharmaceutical and cosmetic industries for evaluation of corrosion, skin irradiation and photo-toxicity [8]. Some of them can also be applied in basic research for clinical use [9].

Human skin gives protective, perceptive and communication functions to the body with resilient and relatively impermeable barrier. To develop agents to deliver pharmacy or active ingredients across skin tissues is a highly attractive topic in recent years. Using compounds or physical equipment to enhance cargo delivery causes some problems, for instance, skin toxicity, skin irritation, inconvenience and high costs [10,11]. Comparing with chemical or physical ways, peptide with cell or skin penetrating activity is an alternative choice. Cell-penetrating peptides (CPP) are peptides that can transport cargos such as chemical compounds, proteins, peptides, and nanoparticles into cells [12]. Most CPP sequences are rich in positively charged residues, and are internalized after interacting with negatively charged glycosaminoglycans (GAGs) and clustering on outer membrane surfaces [13]. For example, a modified PTD (Protein transduction domain) peptide from human immunodeficiency virus (HIV): tat (RKKRRQRRR) has been shown to have cell membrane penetration property and deliver therapeutic proteins into mammalian cells [14]. Another case is AID (arginine-rich intracellular delivery) peptides which successfully enter and deliver functional proteins into epidermis and dermis of mouse [15,16]. Many modified AID peptides (HGH6, TAT, R7) proven to penetrate into skin of living animals with cargos [17,18]. For example, R7-CsA could reach dermal lymphocytes and inhibit cutaneous inflammation [19]. These facts indicated a new approach for increasing delivery of poorly absorbed ingredients across skin tissue barriers.

Here a 10-residue peptide, covering major GAG binding motif of a human ribonuclease, is identified as a CPP_AIF_ (anti-inflammatory CPP). CPP_AIF_ has been shown to possess epithelial cell, GAG and lipid binding properties as well as cell penetrating activity through macropinocytosis [20,21]. Notably, CPP_AIF_ is able to deliver small fluorescent molecules, recombinant proteins, nanoparticles, and peptidomimetic drugs into cells [20]. Based on these facts, safety and potential of CPP_AIF_ for cosmeceutical application were examined with skin cell and 3D-skin models following the Organization for Economic Co-operation and Development (OECD) guidelines with special focus on stability, safety, skin irritation, skin barrier function, *chemico* sensitization, bio-functions and transepidermal activity in this work.

## 2. Materials and Methods

### 2.1. Stability Test of CPP_AIF_ under Different Conditions

All chemicals used in this study were purchased from Sigma-Aldrich (St. Louis, MO, USA). All cell lines were purchased from ATCC (American Type Culture Collection, Manassas, VA, USA). For dry powder stability test, CPP_AIF_ (NYRWRCKNQN with unmodified *N*- and *C*-termini; AIF: anti-inflammation, synthesized by Kelowna International Scientific, Taipei, Taiwan) and was dissolved in water, concentration of 1 mg/mL, and then freeze and dry into powder. Samples were separately incubated in 4 and 25 °C for 1, 3 and 7 days. These tubes were collected and stored at −80 °C. For solution stability test, CPP_AIF_ was dissolved in water to 1 mg/mL then used 0.2 µm filter filted and dispense to 100 µL in each tubes. Samples were separately incubated in specific temperatures, including −20, 4, 30 and 50 °C for 1, 3, 7, 14, 21, 30 and 60 days. These tubes were collected and stored at −80 °C. Then, the remainder of CPP_AIF_ were tested with high-performance liquid chromatography (HPLC) (Waters, Milford, MA, USA) Separation was performed on XBridge C18 column (250 mm × 4.6 mm, particle size 5 µm, Waters). The HPLC condition and program: A buffer is ddH_2_O with 0.1% TFA (trifluoroacetic acid), B buffer is acetonitrile with 0.1% TFA. The flow rate is 1 mL/min and acetonitrile gradient from 10 to 50% in 15 min and the percentage of acetonitrile raise to 100% from 16 to 20 min.

### 2.2. In Vitro Skin Irritation Test (OECD 439)

The 3D reconstructed human epidermis tissue model: SkinEthic™ RHE (SkinEthics, Lyon, France) was used to evaluate whether CPP_AIF_ cause skin irritation [8]. Testing procedure involved topical application of testing article (CPP_AIF_) to surface of epidermis and subsequent assessment of effect on cell viability. All 3D-skin tissues were incubated with growth medium for 2 h and then CPP_AIF_ was add to final 1 mM for 42 min treatment. After treatment, testing substance was washed out by 25 times with 1 mL phosphate buffered saline (PBS) and tissues were further incubated in growth medium for 42 h. After incubation, growth medium was substituted by maintenance medium with 3-(4,5-dimethylthiazol-2-yl)-2,5-diphenyltetrazolium bromide (MTT) agent for 3 h incubation. Next, insert (with tissue) were washed with PBS and air dried. Formazan in tissues was extracted with isopropanol and measured by determining the OD at 570 nm using a microplate spectrophotometer (iMark Microplate Absorbance Reader, Bio-Rad, Hercules, CA, USA). Cell viability of treated models was normalized to the negative control, which was set to 100%.

### 2.3. In Vitro Skin Barrier Function Test (Developed from OECD TG 439)

A test procedure was developed to test effect of CPP_AIF_ on the barrier function on SkinEthic™ reconstructed human epidermis (RHE) based on the relevant procedure mentioned in OECD TG 439. This testing procedure involved topical application of CPP_AIF_ onto surface of the epidermis model for 1 h. After 1 h of exposure, CPP_AIF_ was washed by PBS from the surface followed by application of detergent solution (1% Triton X-100) onto surface of the tissue for another 2 h. Then, washed the detergent solution and AlamarBlue cell viability assay (BUF012B, Bio-Rad, Hercules, CA, USA) was performed for assessment of cell viability. Cell viability of treated models was normalized to the negative control, which was set to 100%.

### 2.4. In Chemico Skin Sensitization: Direct Peptide Reactivity Assay (DPRA) (OECD 442C)

The direct peptide reactivity assay (DPRA) is an *in chemico* method which quantified the remaining concentration of cysteine- or lysine-containing peptide after 24 h incubation with the test chemical at 25 °C. Relative peptide concentration was measured by HPLC (Waters, Milford, MA, USA) with gradient elution and UV detection (220 nm). Cysteine and lysine peptide percent depletion values were calculated for a prediction model (Table 1). This model allowed to classify the test chemical to one of four reactivity classes used to support the discrimination between sensitisers and non-sensitisers. The study is carried out according to the OECD guideline 442C (2015).

### 2.5. Macrophage Inflammation Assay

Macrophage inflammatory assay were carried out to evaluate whether CPP_AIF_ displayed anti-inflammation potential [22,23,24]. In this experiment, macrophage (Raw264.7) cells were seeded in 96-well plates (5 × 10^5^ cells/mL) and allowed to attach overnight. After attachment, cells were incubated with various concentrations of CPP_AIF_ for 1 h and followed by stimulation with 1 μg/mL of LPS (lipopolysaccharide). No LPS-added cells were considered as control groups. After incubation, the amount of TNF-α and IL-6 in the medium were analyzed by enzyme linked immunosorbent assay (# KHC3011 and #EH2IL6, Thermo Fisher, Waltham, MA, USA). Cell viability was measured by AlamarBlue cell viability assay (BUF012B, Bio-Rad, Hercules, CA, USA).

### 2.6. Mast Cell Degranulation Assay

Inhibitory effects on release of β-hexosaminidase in RBL-2H3 (rat-basophilic leukemia cell line) were evaluated by a cell degranulation assay [25]. Briefly, 0.2 mL of 5 × 10^5^ cells/mL RBL-2H3 cells were seed in 24-well plates (in Minimum Essential Medium (MEM) containing 10% Fetal Bovine Serum (FBS), streptomycin (100 µg/mL) and penicillin (100 units/mL)) and sensitized with anti-DNP IgE (0.45 µg/mL). These cells were washed with Siraganian buffer (5 mM KCl, 0.4 mM MgCl_2_, 119 mM NaCl, 25 mM piperazine-*N*,*N* 2-bis(2-ethane sulfonic acid (PIPES), and 40 mM NaOH, pH 7.2) with glucose, 1 mM CaCl_2_, and 0.1% bovine serum albumin (BSA). After washing, these cell were incubated in 160 µL of the PBS for 10 min at 37 °C and then 20 µL of test sample solution or calcimycin (final 1 mM) was added and further incubated for 10 min. The plate was cooling on ice for 10 min to stop reaction. 50 µL of supernatant was transferred to 96-well plate and added with 50 µL of substrate (1 mM *p*-nitrophenyl *N*-acetyl-β-d-glucosaminide) in 0.1 M citrate buffer (pH 4.5) at 37 °C for 2.5 h. The reaction was stopped by adding 200 µL of carbonate-bicarbonate buffer (0.1 M, pH 10.0). The value of A_405_ was measured with a microplate reader.

### 2.7. Internalization of CPP_AIF_ in Human Keratinocyte (HaCaT)

Confocal Laser-scanning Microscopy (CLSM, Zeiss Cell Observer-Z1, Baden-Württemberg, Germany) was performed to assess distribution of TMR-labeled CPP_AIF_ (tetramethylrhodamine-NYRWRCKNQN, synthesized by Kelowna International Scientific, Taipei, Taiwan) in human keratinocytes, HaCaT. First, apply a layer of type I collagen on a glass cover slip, dry and sterilize the cover slip with UV light. Then, culture the HaCaT cells (1 × 103 per slip) on the coverslip for 16 h. The attached HaCaT cells were cultured with 20 μM of the TMR- CPP_AIF_ for 30 min, observed with CLSM (63× oil-immersion objective lens), and photographed under same exposure conditions.

### 2.8. Transepidermal Measurement of CPP_AIF_ by Reconstructed Human Epidermis Tissue Model

3D human epidermis tissue model (SkinEthic™ RHE, SkinEthics, Lyon, France) and two kinds of CPP_AIF_-containing emulsion systems (water-in-oil emulsion (W/O) and oil-in-water emulsion (O/W)) were prepared for the transepidermal measurements. Formulation contents of the W/O and O/W are listed in Appendix A. All 3D-skin tissues were incubated with growth medium for 2 h and then 0.1 mL formulated CPP_AIF_ (0.1 mM) emulsions were add to top of the tissues for 1 h treatment. After treatment liquid beneath the model was collected for HPLC quantification. Transepidermal rate was calculated by measuring the amount of CPP_AIF_ in medium beneath the 3D model by HPLC ((concentration of the beneath liquid/concentration of the topical exposure) × 100).

### 2.9. Statistical Analyses

All statistical analyses were processed by GraphPad Prism version 5.01 for Windows 548 (GraphPad Software, La Jolla, CA, USA). Each value was the average of three measurements, where the presented data was the mean ± SD and all means were compared by one-way ANOVA.

## 3. Results

### 3.1. Stability of CPP_AIF_ under Different Conditions

CPP_AIF_ was incubated at a specified temperature (4 °C and 25 °C) and analyzed remaining quantity by HPLC. The molecular weight of CPP_AIF_ was validated by matrix-assisted laser desorption/ionization-time-of-flight (MALDI-TOF) mass spectrometry. The result indicated that m/z of CPP_AIF_ was 1381.1 as expected (Appendix A). The results in Figure 1 show that CPP_AIF_ maintained intact close to 100% in dry powder form at low temperature (4 °C) and normal temperature (25 °C) up to 1 week, indicating that preservation of CPP_AIF_ in dry powder form could effectively prevent peptide precipitation and fragmentation.

CPP_AIF_ under different solutions or temperatures was further examined to explore suitable storage condition. It was dissolved in ddH_2_O at different temperatures and time to examine which solution was suitable for storage. First, CPP_AIF_ was dissolved in water at a concentration of 1 mg/mL passed through 0.2 µm filter, and incubated at different temperatures. After one week, residual quantity of CPP_AIF_ was still more than 90% when tested at −20 °C and 4 °C, but that of CPP_AIF_ incubated at 30 °C and 50 °C dropped to about 70% (Figure 2). Moreover, after storage for 30 days intact CPP_AIF_ retained in solution was measured to be 81% at −20 °C, 73% at 4 °C, 36% at 30 °C, and 17% at 50 °C, indicating that CPP_AIF_ had better stability at low temperature environment (−20 °C and 4 °C) (Figure 2). MALDI-TOF mass spectrometry was applied to validate the molecular weight of residual CPP_AIF_ as shown in the signal of HPLC chromatogram (Appendix A).

### 3.2. Safety of CPP_AIF_

In vitro skin irritation testz of CPP_AIF_ were carried out following OECD Test Guideline No. 439 (2015), using a reconstructed human epidermis test method.

As shown in Figure 3, relative viability of negative control (NC, PBS), positive control (PC, 5% sodium dodecyl sulphate, SDS), and CPP_AIF_ (1 mM) was determined to be respectively: 100.0 ± 4.8%, 1.44 ± 0.8%, and 90.3 ± 3.2%, clearly suggesting that CPP_AIF_ (1 mM) did not cause skin irritation (no category) according to the classification of OECD 439.

CPP_AIF_ was subsequently applied to in vitro skin barrier function test (developed from OECD TG 439). A normal stratum corneum was multilayered containing essential lipid profile to produce desired functional barrier with robustness to resist rapid penetration of cytotoxic chemicals, e.g., SDS or Triton X-100. Here, a 3D human epidermis tissue model, SkinEthic™ RHE, was used for skin barrier function tests. Tissues incubated with CPP_AIF_ (1 mM) remained cell viability over 80%, which was considered not influencing barrier function of the tissue. As shown in Figure 4, relative viability of negative control (NC, H_2_O), positive control (PC, 5% SDS), and CPP_AIF_ (1 mM) was respectively 100.0 ± 6.2%, 24.0 ± 3.3%, and 95.6 ± 2.8%, evidently indicating that CPP_AIF_ would not impair the skin barrier function of the 3D epidermis tissue model.

Finally, in chemico skin sensitization of CPP_AIF_ was tested following OECD Test Guideline No. 442C. The mean of cysteine and lysine % depletion of 100 mM cinnamaldehyde (positive control), phenoxyethanol, caprylyl glycol, hexalene glycol, 1,3-butanediol and 0.1 mM CPP_AIF_ was respectively calculated to be 65.07, 0.56, 2.53, 0.63, −0.12 and 0.74 (Table 2).

The HPLC chromatograms for cysteine and lysine depletion quantification of CPP_AIF_ were shown in Appendix A. The value of CPP_AIF_ was lower than 19% and thus classified as “Non-sensitizer” according to OECD 442C like other regulatory approved cosmetic ingredients.

### 3.3. Bio-Function of CPP_AIF_: Anti-Inflammation without Sensitization

Inflammation inhibition effect of CPP_AIF_ on macrophage cells was tested at various concentrations of 1 and 0.1 μM. As shown in Figure 5A, 0.1 μM CPP_AIF_ inhibited 22.1 ± 1.2% TNF-α secretion and 1 μM CPP_AIF_ showed stronger inhibition effect of 56.7 ± 2.5%. 0.1 μM and 1 μM CPP_AIF_ also inhibited 18.3 ± 3.4% and 40 ± 4.2% of IL-6 secretion, respectively (Figure 5B). These results implied a dose-dependent relationship between the concentration of CPP_AIF_ (from 0~1 μM) and the amount of TNF-α or IL-6 inhibition. Nevertheless, cell viability remained 85.2 ± 4.25% at CPP_AIF_ concentration of 1 μM. Taken together, CPP_AIF_ might be a potential ingredient with skin protectant function, especially anti-photo aging [26,27] related to the anti-inflammatory effects [28].

The sensitization of CPP_AIF_ was evaluated by mast cell degranulation assay. Here RBL-2H3 cells were treated with various concentrations (0, 0.1, 1, 2, 5, 10 μM) of CPP_AIF_. Granule release represented by β-hexoaminidase was induced by A23187 (Calcimycin) as positive control (PC). In this experiment CPP_AIF_ under 10 μM did not induce any mast cell degranulation release (Figure 6) while cell viability remained over 90%. This result indicated that CPP_AIF_ would not induce any allergic effect in epidermal tissue [29].

### 3.4. Cell Penetration and Transepidermal Test. of CPP_AIF_:

First, epidermal cell penetration activity of CPP_AIF_ was measured in human keratinocyte HaCaT cells. The cells were incubated with 20 µM TMR-CPP_AIF_ at 37 °C for 30 min prior to observation by CLSM (Scale bar: 10 µm). Nuclei were stained with DAPI (4’,6-diamidino-2-phenylindole). As shown in Figure 7, after addition of 20 µM TMR-CPP_AIF_ (tetramethylrhodamine, TMR) for 30 min, TMR signal was clearly detected in the cells (Figure 7). A strong signal accumulation in the cytoplasm showed that TMR-CPP_AIF_ internalized into the cells. Such effect was observed while the skin barrier function still maintained well as investigated by 3D human epidermis tissue model. This result indicated that TMR-CPP_AIF_ could penetrate skin tissue without interupting function of skin tissue.

After confirming the cell penetration activity of CPP_AIF_ to normal human keratinocyte, transepidermal activity of CPP_AIF_ in different formulations was evaluated by 3D reconstructed human epidermis tissue model. After 4 h of exposure the liquid at the bottom of the 3D model was collected and analyzed by HPLC. As shown in Appendix A, CPP_AIF_ appeared as a sharp peak at retention time of 9 min. It was also observed that both emulsion compositions did not influence stability of CPP_AIF_, indicating that CPP_AIF_ could be added to properly designed formulation as a functional cosmetic ingredient.

Transepidermal degree of CPP_AIF_ was further calculated according to HPLC results. ((concentration of CPP_AIF_ in the beneath liquid/concentration of CPP_AIF_ in the topical exposure) * 100). As shown in Figure 8, after 4 h of exposure to the emulsions (100 μL containing 100 μM CPP_AIF_), CPP_AIF_ present at the bottom of the 3D skin tissues W/O and O/W formulations was determined to be respectively 41.4 ± 3.6% and 28.4 ± 1.7% respectively in W/O and O/W formulations. The transepidermal activity of W/O CPP_AIF_ was evidently higher than that of O/W formulation. Taken together, appropriate formulation such as W/O emulsion in this study could evidently facilitate transepidermal penetration of CPP_AIF_.

We further examined skin barrier function of the 3D model after 4 h of exposure to formulated CPP_AIF_. As shown in Figure 9, appropriate W/O formulations promoted CPP_AIF_ penetration into stratum corneum while at the same time maintained the robustness barrier function of the epidermis. The 3D models incubated with W/O CPP_AIF_ showed a higher relative cell viability of 76.5 ± 8.5% than O/W CPP_AIF_ formulation (43.3 ± 7.8%). Relative cell viability of negative control (NC, H_2_O) equaled to 100.0 ± 9.6%. Thus, W/O CPP_AIF_ displayed a better transepidermal degree with slight disruption effect on disruption of skin barrier function.

## 4. Discussion

Since 2000 the application of peptides in cosmeceutical products has rapidly increased, and this trend has sped up research and knowledge of physiological properties of peptides. Now, researchers have identified peptide sequences for penetration into skin layers or different cosmetic activities (e.g., anti-ageing, antioxidant, whitening) [2]. The commercial potential for these bio-functional peptides is high, especially for peptides with excellent stability and no toxicity. Many peptides were reported to have different cosmetic activities, but some clinical study results about these peptides were obtained using formulations containing peptides and other active ingredients. These trials did not clearly differentiate the role of peptide from other actives in the formulation. Hence, the results could not be claimed clearly to be the effect of the peptides for skin benefits [2].

Here CPP_AIF_ was tested with regulatory affair approved methods and clear formulations, therefore the safety datum and anti-inflammatory activity were highly credible. CPP_AIF_ in W/O or O/W formulation could pass through 3D human tissue model. Our CPP_AIF_ showed skin penetration activity without disrupting skin barrier function, and its anti-inflammatory activity might alleviate slight inflammation caused by conventional transepidermal methods [2]. As a CPP, this peptide could also carrier cargos into epidermal cell in skin tissue. Skin tissue is composed by four different layers: stratum corneum, viable epidermis, dermis and subcutaneous connective tissue [30]. This structure efficiently blocks penetration of extraneous molecules in to deeper tissue. It has been reported that TAT can be apply for topical drug-delivery, but high cell penetrating activity might increase some risks which is that TAT might bring drug penetrating cell-layers into deeper tissues [30]. Unlike TAT, our CPPAIF only penetrate into cytosol of epidermal cells without exocytosis property [21], hence it would not be a concern of drug effect. Environmental conditions of skin surface might be tough for bio-molecules (peptide). Cream formulation could provide a stable environment for cosmetic ingredients and remained longer on the skin surface. Some cosmeceutical peptides were also applied in cream formulation [2]. To reduce these challenges that might disrupt stability of CPPAIF, our strategy was applied peptide with W/O or O/W formulation as a mimic of cream in transepidermal test. This standard cosmetic formulation might prove a relatively stable condition for CPPAIF and slightly enhance transepidermal activity. Taken together, CPPAIF itself has been demonstrated to be safe and effective for cosmeceutical use. Comparison between W/O and O/W formulations revealed that the former was more suitable for further application, and the latter with reduced skin barrier function presumably due to formulation components, which might be improved with alternative composition or process. With these facts, CPP_AIF_ was convinced to be a perfect choice for carrying active ingredients through skin tissue and that will be our next goal.

## 5. Conclusions

CPP_AIF_, a GAG binding peptide, could penetrate cell membranes with cargos in living animals and was proven to be stable in powder form under room temperature or in water solution under −20 °C using HPLC analysis. Following OECD guidelines, CPP_AIF_ was evaluated and characterized without skin irritation, skin sensitization and did not disrupt skin barrier function using 3D skin model. In addition to high safety CPP_AIF_ was identified to inhibit inflammation by decreasing inflammatory cytokines, TNF-α and IL-6, using macrophage model. With this bio-function, CPP_AIF_ was further proven to not have sensitization effects. The result of penetration test in HaCaT cells verified CPP_AIF_ for cargo (TMR in this case) delivery into cell in less 30 min. Finally, two commonly used formulations were applied to evaluate transepidermal activity of CPP_AIF_ in 3D skin model in order to imitate real cosmetic applications, and W/O formulation of CPP_AIF_ was identified as a better choice to efficiently penetrate 3D skin with slight disruption of skin barrier function.

## Figures and Tables

**Figure 1 biomolecules-10-00101-f001:**
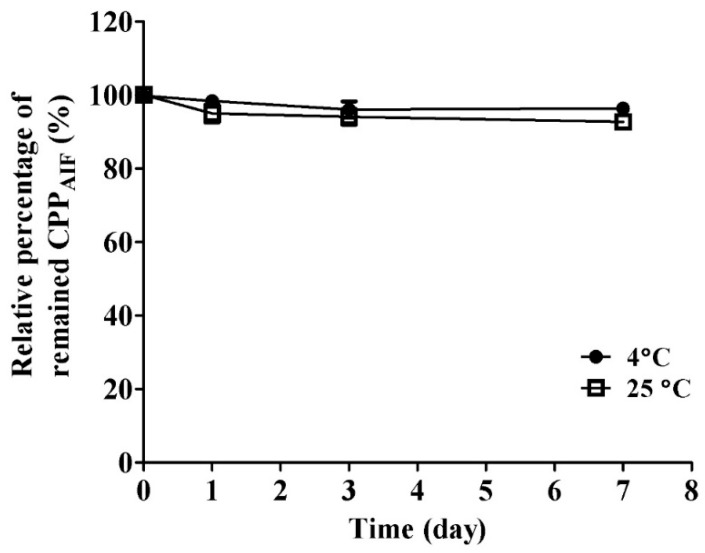
Stability of CPP_AIF_ stored as dry powder at different temperatures. Each sample was dissolved in milliQ H_2_O, and filtered through 0.2 μm filter, then freeze dried into powder form. The samples (100 µL, 1 mg/mL) were separately incubated at 4 °C and 25 °C for 1, 3, and 7 days. The residual amount of CPP_AIF_ ingredient was quantified by HPLC equipped with C18 column.

**Figure 2 biomolecules-10-00101-f002:**
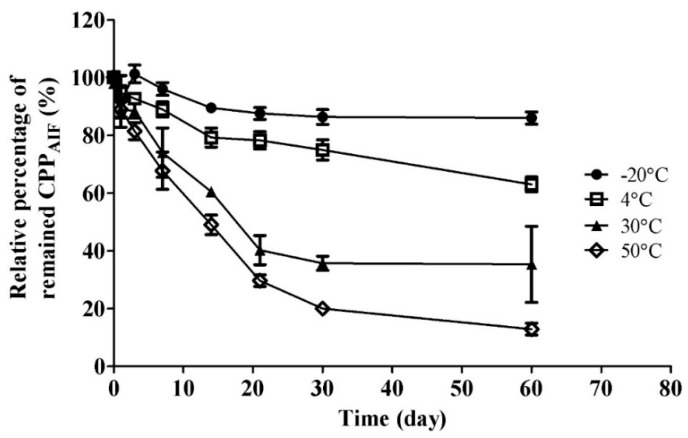
Stability of CPP_AIF_ stored in water at different temperature. Stability of CPP_AIF_ in water at various temperature (−20 °C, 4 °C, 30 °C, 50 °C) for different time duration. Each sample (100 µL, 1 mg/mL) was filtered through 0.2 μm filter, and residual amount of CPP_AIF_ ingredient was quantified by HPLC equipped with C18 column.

**Figure 3 biomolecules-10-00101-f003:**
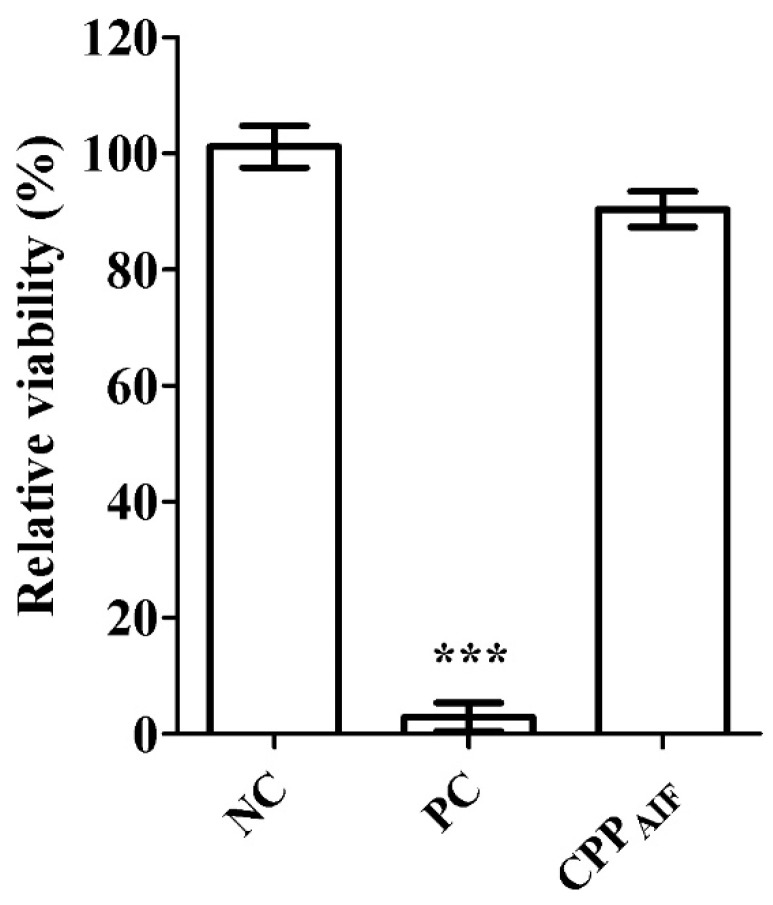
In vitro skin irritation test of CPP_AIF_ in SkinEthic^TM^ RHE model. 3D reconstructed human epidermis tissues were incubated with growth medium for 2 h followed by treatment with NC, PC, and 1 mM CPP_AIF_ for 42 min. Then the tissues were washed and further incubated in growth medium for 42 h. Afterwards, the growth medium was substituted with maintenance medium containing MTT agent for an additional 3 h incubation. Next, the insert (with tissue) was washed with PBS and air dried. Formazan in tissues was extracted with isopropanol and measured by OD at 570 nm. PBS was applied as negative control (NC) and set as 100% (mock), 5% SDS was applied as positive control (PC). *** *p* < 0.001 versus the NC.

**Figure 4 biomolecules-10-00101-f004:**
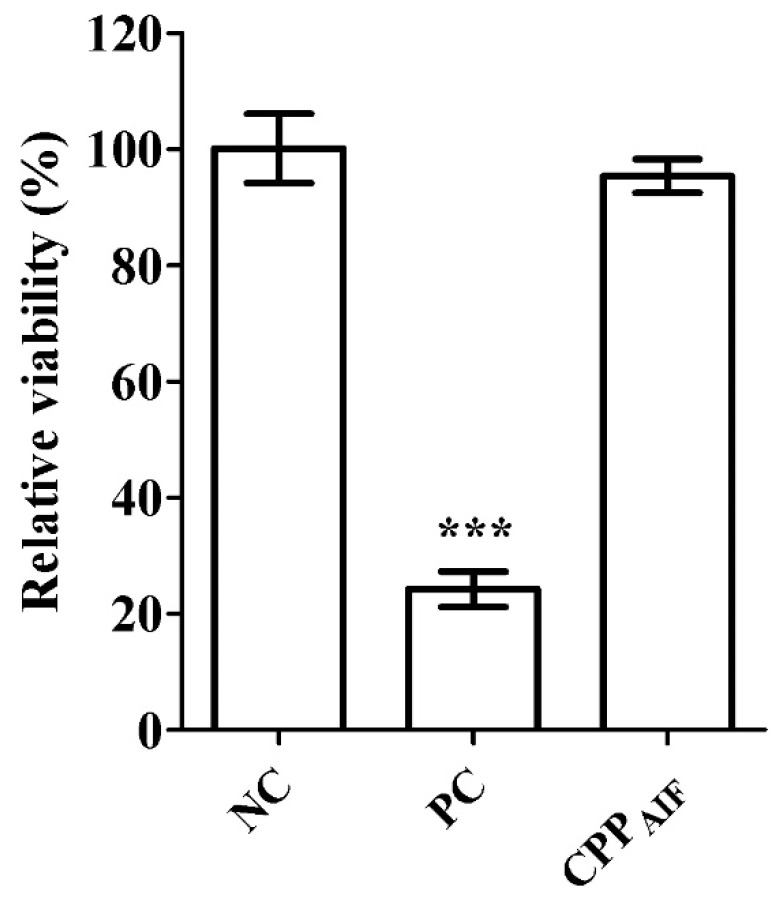
In vitro skin barrier function test of CPP_AIF_ in SkinEthic^TM^ RHE model. 3Dreconstructed human epidermis tissues were exposed with NC, PC, and 1 mM CPP_AIF_ for 1 h. The CPP_AIF_ was washed with PBS from the surface followed by application of detergent solution (1% Triton X-100) onto surface of the tissues for another 2 h. The tissues were washed with PBS and cell viability was measured by AlamarBlue cell viability assay. PBS was applied as negative control (NC) set as 100% (mock) and 5% SDS was applied as positive control (PC). *** *p* < 0.001 versus the NC.

**Figure 5 biomolecules-10-00101-f005:**
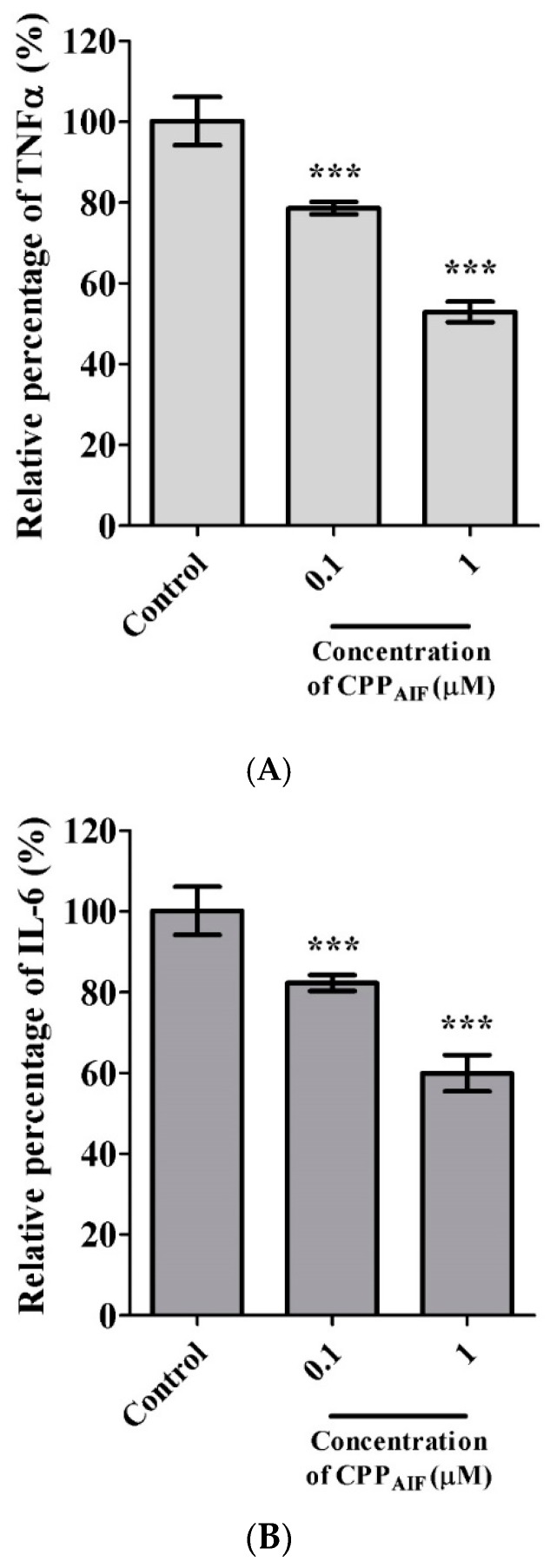
Anti-inflammation activity of CPP_AIF_. Raw264.7 macrophage cells were seeded in 96-well plates (5 × 10^5^ cells/mL) and allowed to attach overnight. After attachment, the cells were incubated with various concentrations of CPP_AIF_ for 1 h followed by stimulation with 1 μg/mL of LPS. The control group was not treated with LPS and its cytokine secretion was set as 100% (mock). The amounts of TNFα (**A**) and IL-6 (**B**) in the medium were analyzed by ELISA. Cell viability was measured by AlamarBlue cell viability assay. *** *p* < 0.001 versus the control group.

**Figure 6 biomolecules-10-00101-f006:**
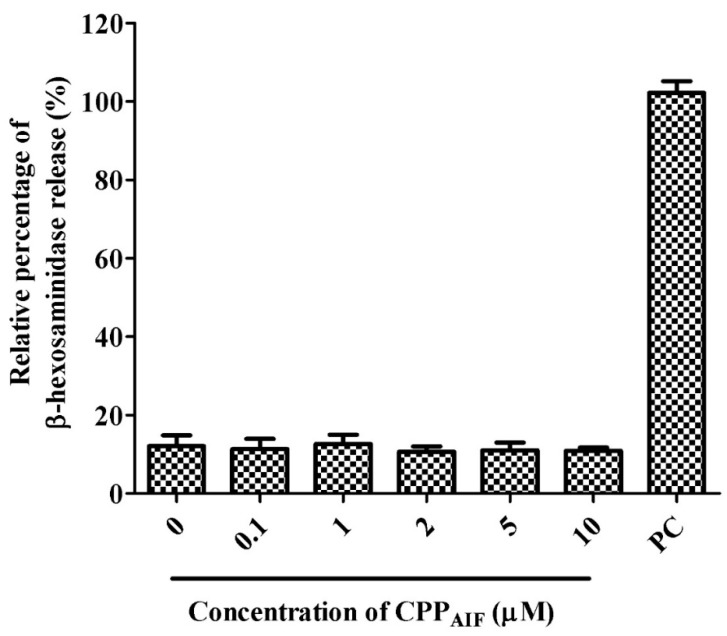
Sensitization test of CPP_AIF_. The sensitization of CPP_AIF_ was evaluated by mast cell degranulation assay. RBL-2H3 mast cells (5 × 10^5^) were treated with various concentrations (0, 0.1, 1, 2, 5, 10 μM) of CPP_AIF_. The positive control (PC) was treated with 1 mM A23187 (calcimycin) and the amount of its granule release was set as 100% (mock).

**Figure 7 biomolecules-10-00101-f007:**
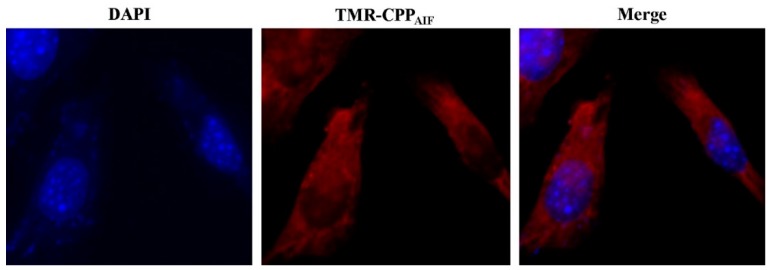
Internalization of TMR-CPP_AIF_. HaCaT cells (1 × 10^3^) were seeded on collagen I coated cover slide and incubated at 37 °C for 16 h. The cells were then incubatd with 20 μM of TMR-CPP_AIF_ at 37 °C for 30 min. After washing and fixing, the cells were mounted and monitored by confocal microscopy. Red: TMR; Blue: nuclear staining with DAPI. (Magnification: 63×).

**Figure 8 biomolecules-10-00101-f008:**
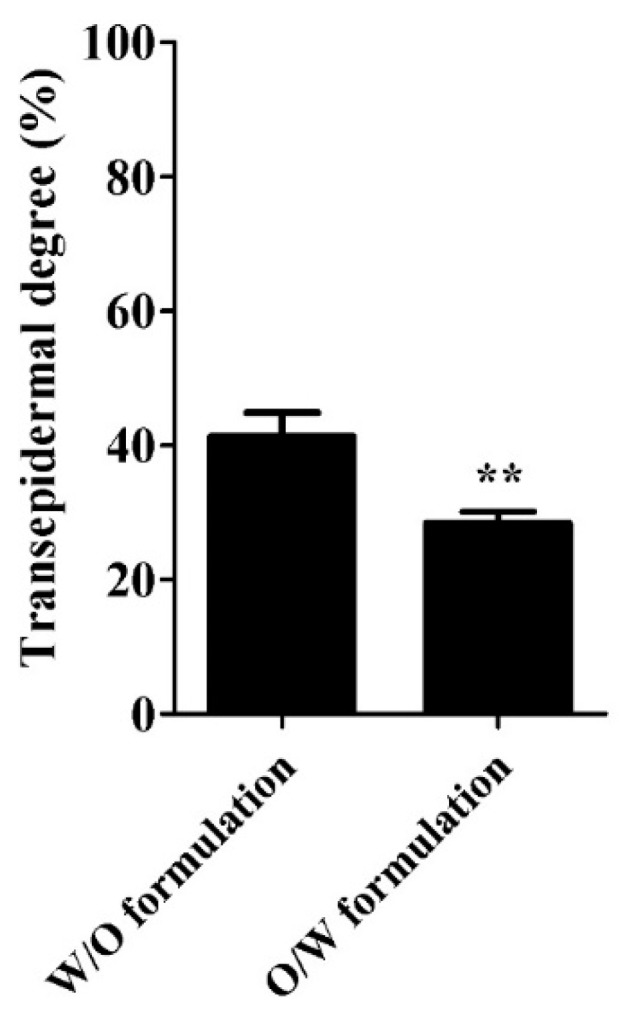
Transepidermal degree of formulated CPP_AIF_ in SkinEthic^TM^ RHE model. All 3D reconstructed human epidermis tissues were incubated with growth medium for 2 h, then 0.1 mL formulated CPP_AIF_ (0.1 mM) emulsions were added to the top of the tissues and incubated for 1 h. After treatment the liquid beneath the 3D skin model was collected for HPLC quantification. Transepidermal degree was calculated by measuring the amount of CPP_AIF_ in the liquid beneath the 3D skin model using HPLC. ** *p* < 0.01 versus the W/O formulation group.

**Figure 9 biomolecules-10-00101-f009:**
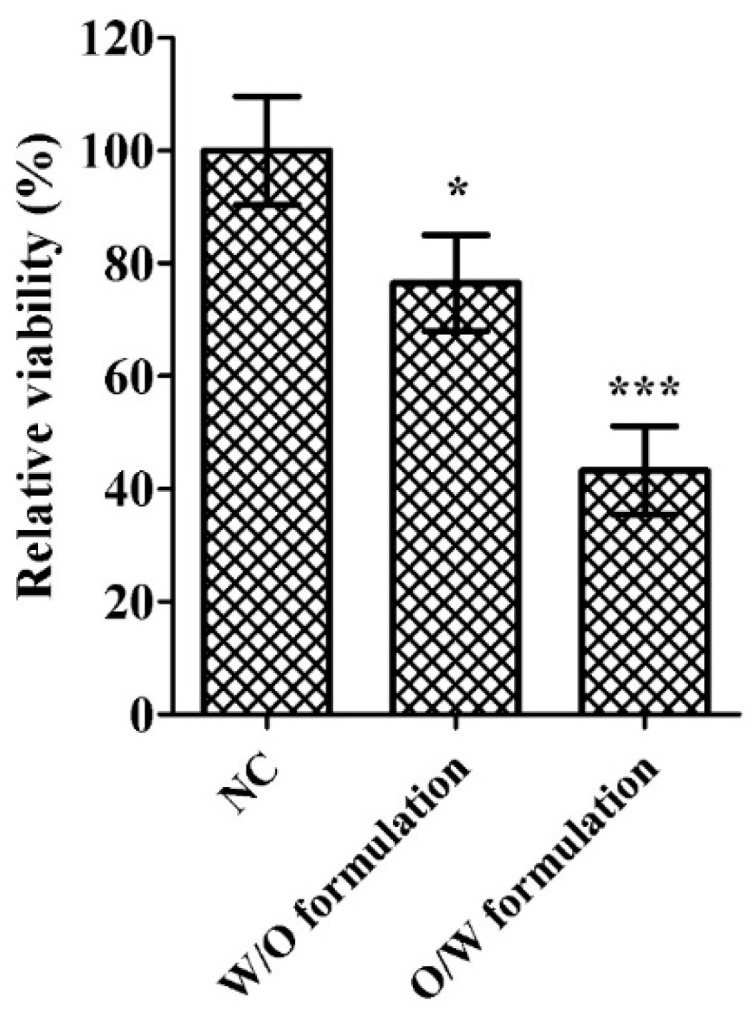
In vitro skin barrier function test of formulated CPP_AIF_ in SkinEthic^TM^ RHE model. 3D reconstructed human epidermis tissues were exposed with formulated CPP_AIF_ (0.1 mM CPP_AIF_) for 1 h. Then CPP_AIF_ was washed by PBS from the surface followed by application of detergent solution (1% Triton X-100) onto surface of the tissues for another 2 h. The tissues were washed and cell viability was measured by AlamarBlue cell viability assay. H_2_O was applied as negative control (NC) in which cell viability was set as 100% (mock). * *p* < 0.05 and *** *p* < 0.001 versus the NC.

**Table 1 biomolecules-10-00101-t001:** Cysteine 1:10/lysine 1:50 prediction model

Mean of Cysteine and Lysine% Depletion	Reactivity Class	DPRA Prediction
0% ≤ mean% depletion ≤ 6.38%	No or minimal reactivity	Negative
6.38% ≤ mean% depletion ≤ 22.62%	Low reactivity	Positive
22.62% ≤ mean% depletion ≤ 42.47%	Moderate reactivity	
42.47% ≤ mean% depletion ≤ 100%	High reactivity	

**Table 2 biomolecules-10-00101-t002:** DPRA calculation according to prediction model.

Test Item	Concentration (mM)	Lysine Depletion	Cysteine Depletion	Mean of Cysteine and Lysine % Depletion	DPRA *^1^ Prediction
Cinnamaldehyde	100	53.82	76.33	65.07	Sensitiser
Phenoxyethanol	100	−0.25	1.38	0.56	Non-sensitiser
Caprylyl glycol	100	2.34	2.72	2.53	Non-sensitiser
Hexalene glycol	100	−0.36	1.62	0.63	Non-sensitiser
1,3-Butanediol	100	−0.36	0.13	−0.12	Non-sensitiser
CPP_AIF_	0.1	1.02	0.45	0.74	Non-sensitiser

*^1^: Direct Peptide Reactivity Assay (OECD Test Guideline No. 442C).

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
