# Peer review of "Cell Penetrating Peptide as a High Safety Anti-Inflammation Ingredient for Cosmetic Applications"

_biomolecules, 2020, doi:10.3390/biom10010101_

Round 1

Reviewer 1 Report

This is an interesting paper that report CPPAIF as a potential candidate for cosmeceutical product development. However before it may be published, there are some issues that have to be addressed as provided below:

Line 18: Change “cosmeceutical peptide” to “cosmeceutical peptides” Line 40: Add a reference at the end of the sentence “....Loren Pickard in 1973.” Line 46: Add reference at the end of the sentence “..... pass skin barrier.” Line 47: Change “tripeptideas” to “tripeptides”. Line 86: The authors should clearly state if the N- and C-terminal of the peptide is either free or modificated. Line 173: the specified temperature values are not in accordance with the temperature values specified in Materials and Methods section (line 88). What information is correct? Line 176: According to Materials and Methods, the tests were performed up to 60 days. What are the results for more than 1 week? Figure 2: Improve figure 2A quality. As it is, it's completely imperceptible. It is preferable to provide figure 2A as supporting information but with proper quality. Line 205: It is the first time that SDS abbreviation appears in the text. According to Biomolecules’ guidelines, abbreviations should be defined in parentheses the first time they appear in the abstract, main text, and in figure or table captions and used consistently thereafter. Most of the abbreviations used in the paper are not defined. Line 239: The abbreviation DRPA is not defined. Table 1: In order to be able to compare CPPAIF peptide with the different controls, the same concentration should be used. In this case, the peptide was tested at a concentration with three orders of magnitutude inferior to the ones used for the controls. The assays of in chemico skin sensitization should be reputed for CPPAIF at a concentration of 100 mM. Line 233: The HPLC chromatograms for cysteine and lysine depletion quantification should be provided as supporting information, at least for the positive control and CPPAIF. Line 273: Change “piror” to “prior”. Line 274: Abbreviation TMR not defined. The authors should also indicate the provenance of the labeled peptide. Statistics are lacking in Figure 3, 4, 5, 8 and 9. Line 276-279: These two sentences are not entirely correct as the 3D human epidermis tissue model was only used for peptide alone and not he labeled TMR-peptide. To state such affirmation the same assay should have been performed for TMR-CPPAIF. The authors should either repeat the assay (more correct) or rephrase. Figure S1: Meaning of W/O and O/W should be specified. Lines 311-312: It is not correct to say that CPPAIF in W/O formulation does not disrupt skin barrier function. This would be true if the bars of the negative control and of W/O formulation would statistically equal in Figure 9. However, as statistics is missing in this figure this cannot be said. Visually, at least, these two bars seem different. Line 361: “without disruption of skin barrier function”. Same as previous comment, it is not to correct to extrapolate to such conclusions. The authors should determine LD 50 value of CPPAIF to HaCat cells to properly assess peptide’s toxicity to this cell line.

Author Response

Thank you for all your comments and suggestions. 
We modified our manuscript according to your comments as below.

Line 18: Change “cosmeceutical peptide” to “cosmeceutical peptides”

Thanks for reviewer’s suggestion.

We have replaced “cosmeceutical peptide” with “cosmeceutical peptides” as shown on page 1 (line #18) in revised manuscript.

Line 40: Add a reference at the end of the sentence “....Loren Pickard in 1973.”

Thanks for reviewer’s suggestion.

We have added reference [3] as instructed on page 1 (line #40) in revised manuscript.

Line 46: Add reference at the end of the sentence “..... pass skin barrier.”

Thanks for reviewer’s suggestion.

We have added reference [4] as instructed on page 1 (line #46) in revised manuscript.

Line 47: Change “tripeptideas” to “tripeptides”.

Thanks for reviewer’s suggestion.

We have replaced “tripeptideas” with “tripeptides” as instructed on page 2 (line #47) in revised manuscript

Line 86: The authors should clearly state if the N- and C-terminal of the peptide is either free or modificated.

Thanks for reviewer’s suggestion.

We have specified “CPPAIF (NYRWRCKNQN with unmodified N- and C-termini)” as instructed on page 3 (line #100) in revised manuscript.

Line 173: the specified temperature values are not in accordance with the temperature values specified in Materials and Methods section (line 88). What information is correct?

Thanks for reviewer’s reminder.

We have corrected temperature values in Materials and Methods section to be “4 °C and 25 °C for 1, 3 and 7 days” on page 3 (line #103) in revised manuscript.

Line 176: According to Materials and Methods, the tests were performed up to 60 days. What are the results for more than 1 week?

Thanks for reviewer’s reminder.

The stability test of dried powder was carried out for 7 days. Because the dried powder was stable, we paid more attention to the stability of CPPAIF in water solution for longer duration (60 day test).

We have corrected temperature values in Materials and Methods section to be to “4 °C and 25 °C for 1, 3 and 7 days” on page 3 (line #103) in revised manuscript.

Figure 2: Improve figure 2A quality. As it is, it's completely imperceptible. It is preferable to provide figure 2A as supporting information but with proper quality.

Thanks for reviewer’s comments and suggestions.

The quality of Figure 2A has been improved to 300 psias shown in Figure S2 in Supplementary data as instructed in revised manuscript.  

Besides, we have added a description about validating molecular weight of CPPAIF by MALDI-TOF mass spectroscopy as following in revised manuscript.
Page 5 (line #191 to 193) “The molecular weight of CPPAIF was validated by MALDI-TOF (Matrix-assisted laser desorption/ionization-time-of-flight) mass spectrometry. The result indicated that m/z of CPPAIF was 1381.1 as expected (Figure S1).”
Page 5 (line#210 to 211)” MALDI-TOF mass spectrometry was applied to validate the molecular weight of residual CPPAIF as shown the signal of HPLC chromatogram  (Figure S2).”

9. Line 205: It is the first time that SDS abbreviation appears in the text. According to Biomolecules’ guidelines, abbreviations should be defined in parentheses the first time they appear in the abstract, main text, and in figure or table captions and used consistently thereafter. Most of the abbreviations used in the paper are not defined.

Thanks for reviewer’s reminder.

We have added definitions of all abbreviations in parentheses the first time they appear in the whole manuscript as instructed.

Line 239: The abbreviation DPRA is not defined.

Thanks for reviewer’s reminder.

We have added “*1: Direct Peptide Reactivity Assay (OECD Test Guideline No. 442C)” to define DPRA in footnote of Table 1 on page 8 (line #256) in revised manuscript.

Table 1: In order to be able to compare CPPAIF peptide with the different controls, the same concentration should be used. In this case, the peptide was tested at a concentration with three orders of magnitutude inferior to the ones used for the controls. The assays of in chemico skin sensitization should be reputed for CPPAIF at a concentration of 100 mM.

Thanks for reviewer’s comments.

The controls at a higher concentration (100 mM) in this test were used to verify non-toxic effect of our platform and demonstrate safety of our CPPAIF. Since the major goal of chemico sensitive test was to ensure the safety of CPPAIF for cosmetic use, we have designed experiments with excess of CPPAIF. The concentration of CPPAIF in this test (0.1 mM) was 10 to 100 times higher than that of cosmetics (0.1 to 10 μM). Our data provide strong evidence of the safety of CPPAIF.

Line 233: The HPLC chromatograms for cysteine and lysine depletion quantification should be provided as supporting information, at least for the positive control and CPPAIF.

Thanks for reviewer’s suggestion.

We have added HPLC chromatograms for cysteine and lysine depletion quantification of CPPAIF in FigureS3 in Supplementary data in revised manuscript.

Line 273: Change “piror” to “prior”.

Thanks for reviewer’s reminder.

We have replaced “piror” with “prior” on page 10 (line #289) in revised manuscript.

Line 274: Abbreviation TMR not defined. The authors should also indicate the provenance of the labeled peptide.

Thanks for reviewer’s reminder

We have added “(Tetramethylrhodamine-NYRWRCKNQN, synthesized by Kelowna International Scientific, Taipei, Taiwan)” in Material and Method section on page 4 (line# 169 to 170) in revised manuscript.

Statistics are lacking in Figure 3, 4, 5, 8 and 9.

Thanks for reviewer’s reminder.

We have added statistical analyses in Figures 3, 4, 5, 8 and 9 on pages 6, 7, 8, 9, 11 and 12 in revised manuscript.

Line 276-279: These two sentences are not entirely correct as the 3D human epidermis tissue model was only used for peptide alone and not the labeled TMR-peptide. To state such affirmation the same assay should have been performed for TMR-CPPAIF. The authors should either repeat the assay (more correct) or rephrase.

Thanks for reviewer’s suggestion

We have replaced “CPP” with “TMR-CPPAIF” on page 9 (line# 295) in revised manuscript.

Figure S1: Meaning of W/O and O/W should be specified.

Thanks for reviewer’s suggestion.

We have specified “W/O: CPPAIF in W/O formulation; O/W: CPPAIF in O/W formulation.” in legend of Figure S4 (S1) as shown in Supplementary data in revised manuscript.

Lines 311-312: It is not correct to say that CPPAIF in W/O formulation does not disrupt skin barrier function. This would be true if the bars of the negative control and of W/O formulation would statistically equal in Figure 9. However, as statistics is missing in this figure this cannot be said. Visually, at least, these two bars seem different.

Thanks for reviewer’s reminder.

We have replaced “without disruption” with “with slight disruption effect on” on page 11 (line# 328) in revised manuscript. 

Line 361: “without disruption of skin barrier function”. Same as previous comment, it is not to correct to extrapolate to such conclusions. The authors should determine LD 50 value of CPPAIF to HaCat cells to properly assess peptide’s toxicity to this cell line.

Thanks for reviewer’s comments.
We have replaced “without” with “with slight” on page 12 (line# 378??) in revised manuscript. 

We have added cytotoxicity test of CPPAIF to HaCaT cell line in FigureS5. CPPAIF (0.1 to 10 μM) has no cytotoxicity to HaCaT cells. In Figure4, in vitro skin barrier function test of CPPAIF on SkinEthicTM RHE model also indicates that CPPAIF has no cytotoxicity to SkinEthicTM RHE model. Combining these data, we suggest that the slight disruption effect of CPPAIF in W/O formulation is presumably correlated with formulation process rather than CPPAIF alone.

Reviewer 2 Report

This research details the examination of a cell penetrating peptide for its potential application as a cosmeceutical. The authors claim that the peptide investigated with the sequence NYRWRCKNQN and denoted as CPPAIF posses anti-inflammatory property and could transport cargo into epithelial cells without disrupting the skin barrier when evaluated using a 3D skin model when administered in water or oil formulations. 

Various tests were conducted to analyse the efficacy and safety of the peptide, including:

Stability test by dissolving the peptide in water and incubating at different temperatures followed by HPLC analysis. Any detrimental interaction of the peptide with a 3D reconstructed human epidermis tissue model was performed by topical application of the CPPAIF to the epidermis followed by MTT assay to measure cell viability. Anti-inflammatory properties of the CPPAIF were evaluated via a macrophage inflammation assay by stimulating the macrophages with an LPS insult followed by treatment with CPPAIF and then measuring the amount of TNF-α and IL-6 using ELISA. Mast cell degranulation assay was further performed to investigate the allergic effect of the CPPAIF on the epidermal tissue. Skin penetration activity of CPPAIF was measured in human keratinocyte.   

The introduction briefly mentions cosmeceutical peptides that are currently on the market but there is no mention of how these cosmeceutics penetrate the stratum corneum and whether there are other cell penetrating peptides that have been previously investigated for this purpose. Also the introduction briefly dives straight into the use cell penetrating peptides which are classically used to deliver large macromolecules, recombinant proteins and peptides into cells in an in vivo setting rather than for topical applications.   

Some of the discussion which mentions the use of cell penetrating peptides like the modified TAT sequence for delivery of cargo across skin tissue should be moved to the introduction.

With regards to the research design the rationale is not clear as cell penetrating peptides are primarily used to deliver a therapeutic cargo as further exemplified in the discussion. However, the peptide tested had no therapeutic cargo attached to it. Do the authors intend to load the CPP with  a macromolecular cosmeceutic to be delivered through the skin and if so wouldn't it be more prudent to perform all the above mentioned analyses on the CPP-cargo conjugate rather than the CPP itself.    

Again in the discussion the authors mention, based on their results, the ability of the peptide to permeate and enhance concentration of active  ingredients in the dermis. This is still speculative and cannot be ascertained by simply observing the internalisation of the CPP conjugated to a tmr fluorophore.

There is also a requirement where the penetration of cosmeceuticals within the skin should not be deep enough to classify it as a drug which should be addressed in the discussion. 

The authors detail the stability of the CPPAIF in solution at different temperatures and show that the peptide has an increased shelf life at lower temperatures. However there is no study done to show the stability of this CPP to proteolytic enzymes in human eccrine sweat once applied topically and how that would affect the efficacy of the CPP in delivering the active ingredient.

The oil-in-water emulsion (o/w) formulation of the CPP should slightly reduced transepidermal activity but a significantly reduced realtive cell viability when compared to the water-in-oil emulsion (w/o) formulation. This should be further discussed.

The discussion and conclusion sections are succinct and should be more elaborate taking into account the above mentioned concerns. 

Author Response

Thank you for all your comments and suggestions.
We modify our manuscript as below

1. The introduction briefly mentions cosmeceutical peptides that are currently on the market but there is no mention of how these cosmeceutics penetrate the stratum corneum and whether there are other cell penetrating peptides that have been previously investigated for this purpose. Also the introduction briefly dives straight into the use cell penetrating peptides which are classically used to deliver large macromolecules, recombinant proteins and peptides into cells in an in vivosetting rather than for topical applications.  Some of the discussion which mentions the use of cell penetrating peptides like the modified TAT sequence for delivery of cargo across skin tissue should be moved to the introduction.

Thanks for reviewer’s comments and suggestions.

We have shifted the description about cell penetration peptide, TAT and AID, from discussion section to the introduction section as shown on page 1 (lines # 75 to 95) in revised manuscript.

2. With regards to the research design the rationale is not clear as cell penetrating peptides are primarily used to deliver a therapeutic cargo as further exemplified in the discussion. However, the peptide tested had no therapeutic cargo attached to it. Do the authors intend to load the CPP with a macromolecular cosmeceutic to be delivered through the skin and if so wouldn't it be more prudent to perform all the above mentioned analyses on the CPP-cargo conjugate rather than the CPP itself.   Again in the discussion the authors mention, based on their results, the ability of the peptide to permeate and enhance concentration of active ingredients in the dermis. This is still speculative and cannot be ascertained by simply observing the internalisation of the CPP conjugated to a tmr fluorophore.

Thanks for reviewer’s comments.

In our design CPPAIF itself can act as an anti-inflammatory peptide and can also transport cargo pass through the skin barrier. As CPPAIF is known to carry cargo into cells (reference 20), in this study we used TMR-CPPAIF with red fluorescence for easy observation of cell penetrating activity of CPPAIF. Indeed, we have demonstrated that CPPAIF can penetrate into normal human keratinocyte and the distribution in cells is similar to that of previous report (reference 20) as expected.

For transepidermal test which is a simulation of cosmetic application, we use CPPAIF itself in different formulations and quantify amount of CPPAIF passing through 3-D skin tissue model by HPLC.   

We have replaced “Transepidermal test of CPPAIF “ with “Cell penetration and transepidermal test of CPPAIF” in Result section on page 9 (lines# 287), replace “Skin penetration “ with “First ,epidermal cell penetration” in Result section on page 9 (lines# 288) and add ” After confirming the cell penetration activity of CPPAIF to normal human keratinocyte,” in Result section on page 10 (lines# 301) in revised manuscript.

Indeed, “cell penetration” and “skin tissue penetration” are two different functionsm which have been clearly described in Result section 3.4

Thus, we have deleted

“CPPAIF might also solve one big problem for cosmetic area: “how to surmount the skin barrier to permeate and enhance concentration of active ingredients in the dermis” in Discussion section in revised manuscript.

We have rewrited “As a CPP, this peptide showed extremely high skin penetration activity without disrupting skin barrier function and its anti-inflammatory activity could inhibit unexpected effects caused by slight inflammation.” as “CPPAIF in W/O or O/W formulation could pass through 3D human tissue model. Our CPPAIF showed skin penetration activity without disrupting skin barrier function, and its anti-inflammatory activity might alleviate slight inflammation caused by conventional transepidermal methods [2].” in Discussion section on page 12 (lines# 348 to 351) in revised manuscript.

3.There is also a requirement where the penetration of cosmeceuticals within the skin should not be deep enough to classify it as a drug which should be addressed in the discussion. 

Thanks for reviewer’s reminder.

Previous report (reference 21) indicates that CPPAIF only penetrates into cells and accumulates in cytoplasm. This result indicates that CPPAIF (with cargo or not) might not penetrate into deeper tissue of skin.

We have added

“Skin tissue is composed by four different layers: stratum corneum, viable epidermis, dermis and subcutaneous connective tissue [30]. This structure efficiently blocks penetration of extraneous molecules in to deeper tissue. It has been reported that TAT can be apply for topical drug-delivery, but high cell penetrating activity might increase some risks which is that TAT might bring drug penetrating cell-layers into deeper tissues [30]. Unlike TAT, our CPPAIF only penetrate into cytosol of epidermal cells without exocytosis property [21], hence it would not be a concern of drug effect.” in Discussion section on page 12 (lines# 352 to 358) in revised manuscript.

4. The authors detail the stability of the CPPAIF in solution at different temperatures and show that the peptide has an increased shelf life at lower temperatures. However there is no study done to show the stability of this CPP to proteolytic enzymes in human eccrine sweat once applied topically and how that would affect the efficacy of the CPP in delivering the active ingredient.

Thanks for reviewer’s comments.

We have applied our CPPAIF with W/O or O/W formulation following standard cosmetic procedure for cream to provide a relatively stable condition for CPPAIF.
We have added

“Environmental conditions of skin surface might be tough for bio-molecules (peptide). Cream formulation could provide a stable environment for cosmetic ingredients and remained longer on the skin surface. Some cosmeceutical peptides were also applied in cream formulation[2]. To reduce these challenges that might disrupt stability of CPPAIF, our strategy was applied peptide with W/O or O/W formulation as a mimic of cream in transepidermal test. This standard cosmetic formulation might prove a relatively stable condition for CPPAIF and slightly enhance transepidermal activity.”

in Discussion section on page 12 (lines# 358 to 364) in revised manuscript.

5. The oil-in-water emulsion (o/w) formulation of the CPP should slightly reduce transepidermal activity but a significantly reduced realtive cell viability when compared to the water-in-oil emulsion (w/o) formulation. This should be further discussed.

Thanks for reviewer’s reminder.

We have added

“Taken together, CPPAIF itself has been demonstrated to be safe and effective for cosmeceutical use. Comparison between W/O and O/W formulations revealed that the former was more suitable for further application, and the latter with reduced skin barrier function presumably due to formulation components, which might be improved with alternative composition or process.”

in Discussion section on page 12 (lines# 364 to 368) in revised manuscript.

Round 2

Reviewer 2 Report

I am satisfied with the revisions.